# How Reliable Are Ultra-Short-Term HRV Measurements during Cognitively Demanding Tasks?

**DOI:** 10.3390/s22176528

**Published:** 2022-08-30

**Authors:** André Bernardes, Ricardo Couceiro, Júlio Medeiros, Jorge Henriques, César Teixeira, Marco Simões, João Durães, Raul Barbosa, Henrique Madeira, Paulo Carvalho

**Affiliations:** Centre for Informatics and Systems of the University of Coimbra (CISUC), 3030-290 Coimbra, Portugal

**Keywords:** ultra-short-term HRV features, statistical significance, correlation, cognitively demanding tasks, code comprehension

## Abstract

Ultra-short-term HRV features assess minor autonomous nervous system variations such as variations resulting from cognitive stress peaks during demanding tasks. Several studies compare ultra-short-term and short-term HRV measurements to investigate their reliability. However, existing experiments are conducted in low cognitively demanding environments. In this paper, we propose to evaluate these measurements’ reliability under cognitively demanding tasks using a near real-life setting. For this purpose, we selected 31 HRV features, extracted from data collected from 21 programmers performing code comprehension, and compared them across 18 different time frames, ranging from 3 min to 10 s. Statistical significance and correlation tests were performed between the features extracted using the larger window (3 min) and the same features extracted with the other 17 time frames. We paired these analyses with Bland–Altman plots to inspect how the extraction window size affects the HRV features. The main results show 13 features that presented at least 50% correlation when using 60-second windows. The HF and mNN features achieved around 50% correlation using a 30-second window. The 30-second window was the smallest time frame considered to have reliable measurements. Furthermore, the mNN feature proved to be quite robust to the shortening of the time resolution.

## 1. Introduction

Heart rate (HR) is defined as “the number of heartbeats per minute” [1], and does not provide direct information about autonomic nervous system (ANS) dynamics since it is a static index of autonomic input to the sinoatrial node [2]. On the other hand, heart rate variability (HRV) is described as “the fluctuation in the time intervals between adjacent heartbeats” (RR intervals or NN intervals) [1]. RR intervals are measured in milliseconds (ms) and “result mostly from the dynamic interaction between the parasympathetic and the sympathetic inputs to the heart through the sinoatrial node” [3]. Unlike HR, HRV analysis is useful for providing a “quantitative assessment of cardiac autonomic regulation” [2]. 

HRV is often used as a non-invasive marker of ANS activity, and its spectrum analysis can measure the sympathovagal balance [4]. Thereby, many studies point to the potential of HRV for diagnosis and prognosis of health problems [5] and other areas such as the measurement of a subject’s cognitive load [6,7,8].

From the physiological point of view, our nervous system has two major subdivisions: the central nervous system (CNS) and the peripheral nervous system (PNS). The CNS is formed by the spinal cord and the brain and is responsible for receiving signals from different body components, processing these signals and producing responses as new signals to be delivered across these body components. The PNS is the subdivision accountable for carrying messages exchanged between the CNS and the organs, glands, muscles and senses [9]. Belonging to the PNS, we have a further two different constituents: the somatic nervous system (SNS), which is the constituent responsible (essentially) for voluntary and conscious actions, and the autonomic nervous system (ANS), where our study focuses. The ANS has two subdivisions: the sympathetic and the parasympathetic nervous systems. The sympathetic nervous system is more associated with stressful situations that need an emergency response, the so-called fight-or-flight mode.

In contrast, parasympathetic nervous system activity is associated with more conservative and restoring processes, bringing the body back to a stable state [6]. In this way, the CNS has the power to influence every other system in the body, which, in theory, makes it possible to access data, such as the cognitive load of the brain through physiological signals controlled by the ANS, that might be captured using HRV. In fact, HRV is considered an index of autonomic control of the heart and has been pointed out as having a good physiological correlation with cognitive functioning [3]. However, despite several studies in the field, this is still a controversial subject across the scientific community. 

Taking a leading role in the regulation of cardiac function, the ANS controls the constriction and relaxation of blood vessels, which allows it to regulate blood pressure and, in this way, is capable of adjusting heart rate and heart contractility. Using an ECG signal, it is possible to compute HRV through time fluctuations of the intervals between the consecutive R-peaks. Then, from the HRV measurement, different features can be extracted in time, geometric, non-linear and frequency domains, which can be used to access the different ANS sympathetic and parasympathetic systems’ dynamics. Across different papers that approach HRV analysis, the most common features referenced in the time domain are mNN, SDNN, SDSD, RMSSD, NN50 and pNN50 (see terminology in Table 1). Regarding power spectrum density analysis, the frequency domain is divided into three bands: the very-low-frequency band (VLF: under 0.04 HZ), the low-frequency band (LF: 0.04 to 0.15 HZ) and the high-frequency band (0.15 to 0.4 HZ) [10]. The features extracted from each band most referenced in the literature are the total power and the peak. The ratio between the LF power and the HF power is also frequently mentioned.

From the features previously mentioned, some features have already been linked to physiological dynamics. Starting with the VLF band, this band is mentioned to be a heart’s intrinsic nervous system consequence. The SDNN, as mentioned in [10], is influenced by every cyclic component responsible for variability in the recording period. This feature is highly correlated with the LF band, and the two are associated with both the sympathetic and parasympathetic systems’ dynamics. The LF band is as well-linked to blood pressure regulation via baroreceptors. The features RMSSD, pNN50 and the HF band are also correlated and are closely influenced by the parasympathetic system. Thus, the ratio between the LF power and the HF power is believed to be a good measure of the balance between the sympathetic and parasympathetic systems. Although this belief is not consensual, and this relationship is not as straightforward as once believed, we can still look at this ratio as a metric of one system’s predominance over another [7].

In addition to time and frequency domains, several authors have also pursued the extraction of measures in the non-linear space in order to unveil non-linear HRV patterns. Based on the studies present in the literature, several measurements have been selected, focusing on their consistency when extracted using small time frames (e.g., 5 min) [2,6,11,12], which are: approximate entropy, Poincare’s plot parameters (SD1 and SD2), point transition measure, Katz fractal dimension and Higuchi fractal dimension from the non-linear domain, stress index, HRV triangular index and TINN from the geometric domain.

Short-term time frames (5 min in length) are already a standard and are currently well-accepted as suitable for extracting accurate HRV measurements [10]. However, the need to extract HRV measurements using time frames shorter than 1 min (ultra-short HRV features) has grown for several reasons [5,11,13]. Among these are the need to reduce the time spent and costs in the extraction of these indexes, the fact that they are incompatible with the dynamics of the physiological mechanisms to be captured (e.g., cognitive load spikes), or the need to extract these features in new environments using modern wearable devices [13]. In fact, these brought a wide range of new applications that can benefit from the advances in ultra-short-term measurements fields. 

The current study is integrated into the Biofeedback Augmented Software Engineering (BASE) project, which aims to develop a solution capable of using biofeedback from the programmer to detect code areas more prone to error and save time in the bug detection process. Code comprehension and bug detection tasks consume up to 70% of the programmers working time, representing millions spent every year trying to avoid these software faults [14]. The BASE project aims to solve this significant socioeconomic problem. The interest in using HRV in software engineering is growing very fast, and applications such as the identification of problematic code areas (that may have bugs and need revision) require very fast response in assessing programmers’ cognitive loads using HRV [15]. In order to ensure a real-time response and to detect acute cognitive stress changes, we need time analysis windows as short as possible to achieve the required time resolution, which is the motivation behind the current study.

In order to find out which features are adequate to be used in real-life applications, such as the mentioned code inspection context, and to understand their time frame limitations, this study aims to analyze the stability of 31 existing HRV features. To this end, these features were extracted using time frames of variable sizes (e.g., as low as 10 s) in a code inspection context, an environment very demanding from the intellectual perspective and, therefore, with individuals subjected to high cognitive loads. Our current work investigates the smallest time frame where each feature is reliable, performing an intra-subject and intra-run analysis to avoid biasing the results.

## 2. Background and Related Work

Long-term (recordings lasting 24 h) and short-term (recordings lasting 5 min) HRV measurements are well-documented and conventionally accepted as valid HRV measurements, having multiple clinical applications [10]. However, as mentioned earlier, there is a growing need to use shorter segments. Some studies focus on investigating the reliability of these ultra-short-term measurements when compared to short-term HRV measurements.

In order to evaluate ultra-short-term HRV measurements’ reliability as a surrogate of the short-term HRV, different analyses can be performed. A procedure proposed by Pechia et al. [2] included a correlation analysis to test the existence of a significant association between features. If the correlation was significant and the correlation coefficient was above 0.07, the researchers performed a Bland–Altman plot to analyze the degree of bias. In case the data dispersion remained within the 95% line of agreement, the final step was to perform an effect size statistic (Cohen’s d statistic to parametric data or Cliff’s delta statistic to non-parametric data). The feature was then considered a good surrogate if the effect size statistic test only detected minor differences. The mentioned procedure agrees with Shaffer et al. [5], which recommend using correlation/regression analyses paired with a Bland–Altman plot. Both works agree that only a correlation analysis is not enough to determine if an ultra-short-term HRV feature is a good surrogate of short-term HRV. In fact, the two compared measurements can be highly correlated but still have significantly different values.

A 2017 study carried out by Castaldo et al. [11] used Bland–Altman plots and Spearman’s rank correlation analysis to assess which ultra-short-term HRV features are a valid surrogate of short-term HRV. The study also built a machine learning model using ultra-short-term HRV features to discriminate between stress and rest states. The conclusions were that mean HR, the standard deviation of HR, mNN, SDNN, HF and SD2 are appropriate short-term HRV surrogates for mental stress assessment. The paper also highlighted a machine learning model obtained using the mNN, the standard deviation of HR and the HF features, which achieved an accuracy above 88%.

In an article by Salahuddin et al. [8], the authors used mobile-derived ECG recording to extract several HRV measurements and the Kruskal–Wallis test to analyze the reliability of these measurements. It was “assumed that short-term analysis was not significantly different to the 150-second analysis if the *p*-value was greater than 0.05”, and the goal was to find until which window span a feature is a good estimative of the 150-second window. The authors concluded that mean RR and RMSSD extracted using 10-second windows were not significantly different from the estimates using 150-second windows. This finding was also confirmed when using 20-second windows for extracting pNN50, HF, LF/HF, LFnu and HFnu features, 30-second windows for LF features and 50-second windows for VLF features. As for the remaining features studied by the authors, a minimum time frame of 60 s was necessary for extracting features that were not significantly different from the 150-second reference features. This study’s data were recorded during the subject’s day-to-day activities such as regular daily work, study, physical activities and sleep.

In the work of Baek et al. [16], a similar approach has been used to evaluate the reliability of ultra-short-term HRV measurements as short-term (5 min) HRV surrogates. The data were acquired in 5-minute recordings while the subjects were “sitting at rest in a comfortable chair”. In order to accomplish the proposed goal, the authors computed the *p*-value by the Kruskal–Wallis test, the Pearson correlation r and Bland–Altman plot analysis comparing 5-minute short-term measurements with ultra-short-term ones with different time frames (270, 240,210, 180, 150, 120, 90, 60, 30, 20 and 10 s). The highlighted features with the best results in this study were the mean HR, where 10-second windows were used to obtain results comparable to the 5-minute analysis, the HF, which required 20-second windows, and the RMSSD, which required 30-second windows.

Following similar approaches, other works, such as the publications by Landreani et al. [13], Li et al. [17], Salahuddin et al. [18], Nussinovitch et al. [19] and McNames et al. [20], were able to converge on a common set of conclusions, where mean HR, mean RR, SDSD, RMSSD, pNN50, HF, LF/HF, LFnu and HFnu were shown to be reliable under the 60-second recordings.

From the reported works, it is possible to conclude that ultra-short-term measurements are far from being consensual. Due to their extraction, only some features keep their stability under small window constraints. Additionally, it is still unclear what the time frame limit is for each HRV feature that can be applied to compute a reliable surrogate of its counterpart extracted from 5-minute recordings. Furthermore, the studies found related to this topic were developed with the subjects at rest or performing elementary tasks in controlled environments. In this work, we aim to elucidate these aspects and validate them under stressful and intellectually demanding environments; more precisely, with the subjects performing software code inspection tasks (i.e., bug detection), which is a highly complex, dynamic, and cognitively demanding task. The main goal of our work is precisely to investigate ultra-short-term HRV features to determine whether HRV-based tools can effectively be used in software development environments. To this extent, our present study investigates the smallest time frame, i.e., the finest time resolution, where each feature is reliable, i.e., the smallest time frame where each feature behavior is still representative of the corresponding 180-second measurement, under our experiment context.

Another relevant aspect that is worth mentioning is that the existing studies perform an inter-subject analysis of the features, i.e., perform the correlation or statistical analysis after concatenating the features collected from different subjects. This fact can lead to biased correlation values since it captures the inter-subject feature tendencies that may overwhelm the actual feature tendencies. In order to avoid this kind of bias, our study performs an intra-subject and intra-run feature analysis.

## 3. Methods

### 3.1. Participants

The data used in the current work were collected in the scope of the BASE project and aimed at the research of error making and error discovery during software inspection tasks, using functional magnetic resonance imaging (fMRI) and other non-invasive sensors such as the ECG. In order to collect the data used in the study, we opened a call for participation in the experiment. Through this process, we obtained 49 candidates consisting of a mixture of students (pursuing PhDs and MSs in different computer science fields), academic professors and professional specialists in the software sector (code reviewers). The candidates were then interviewed and screened to guarantee their fitment to the study objectives. During the interview, demographic and biometric characteristics (e.g., age), professional status, programming experience, availability and motivation were collected. Subsequently, each candidate’s proficiency level was also assessed based on the score provided by two questionnaires: (1) a programming experience questionnaire and (2) a technical questionnaire (see Appendix A). The first questionnaire aimed to assess the candidate’s programming experience based on the candidate’s coding volume in the last three years. The second questionnaire’s goal, composed of 10 questions, was to assess the candidate’s coding skills. The programming experience gave us an overall idea of the experience in the past years from the candidate: (1) experience in SW programming (number of years); (2) lines programmed in any language in the last 3 years (approximate number); (3) lines programmed in C in the last 3 years (approximate number); and (4) lines written in the biggest C program written (approximate number). On the other hand, the technical questionnaire was used for candidate characterization regarding present knowledge and coding skills, which is, therefore, more helpful in selecting and classifying the candidates. Based on the results obtained in these questionnaires, the candidates with a score below 3 (out of 10) were considered not eligible since they were not representative of software industry professionals. The remaining ones were characterized as non-experienced (score between 4 and 7) and experienced (score between 8 and 10). In summary, 21 male subjects, ranging from 19 to 40 years, with a median of 22 years, were selected for the experiments after the screening process. 

All subjects provided written informed consent, and all the data were anonymized. This study was approved by the Ethical Committee of the Faculty of Medicine of the University of Coimbra, following the Declaration of Helsinki and the standard procedures for studies involving human subjects.

### 3.2. Experimental Protocol and Setup

The selected candidates were submitted to 4 different runs of code inspection tasks using 4 code snippets written in C code language (selected randomly at each run). Each run started with a fixation cross in the middle of the screen for 30 s. Subsequently, three tasks were presented to the subject: a natural language reading (literary excerpt) task, a neutral (bug-free and straightforward code) code reading task, and one code inspection (code with bugs) task. The order of the presentation was randomly selected to avoid biasing the results, following a randomized control crossover design. Between each task and at the end of each run, a fixation cross was presented to the subject for 30 s. The description of each task is provided as follows:

1. **Natural language reading**: In this task, a text in natural language is presented to the subject (selected randomly from the set of 4 different texts) for 60 s. The presented texts were selected in order to have neutral characteristics and avoid measurement fluctuations induced by narrative-triggered emotions;

2. **Simple code snippet reading**: In this task, the subject is presented with a simple and iterative code snippet (selected randomly from the set of 4 different neutral code snippets) for 300 s. The presented code snippets were selected with the objective of inducing the subject into a state of low cognitive effort which will be used as a reference state during the posterior analysis;

3. **Code inspection**: In this task, a code snippet in C language is displayed to the subject (selected randomly from a set of 4 different code snippets of different complexities) for a maximum of 600 s. In this task, the subject is asked to analyze and inspect the code, aiming for bug detection. 

The schematic representation of each run is provided below (see Figure 1).

Each run lasted about 21 min, meaning the whole protocol lasted about 1 h and 20 min. During the experiment, the subjects were alone in a quiet, isolated room when performing the tasks. Furthermore, the subjects were informed a priori about all the protocol and processes of the experiment, and they were also instructed not to take anything that could stimulate/inhibit them the day before the experiment. The code inspection tasks were presented to participants using the Vizard software [21].

The equipment used to collect the electrocardiogram (ECG) signal was the Maglink RT (Neuroscan) with a sampling frequency of 10 kHz [22] (see equipment set-up in Figure 2). For ECG signal acquisition, electrodes from the Neuroscan equipment were positioned in the V1 and V2 locations. The electroencephalogram (EEG) was also collected using an EEG cap with 64 channels, although the measured biosignal was not used in the current analysis.

### 3.3. Pre-Processing and ECG Segmentation

Given the nature of the experiment, an initial pre-processing was necessary to remove the gradient artifact (GA) induced by the MRI scanner on the ECG signals. To this end, an average artifact subtraction (AAS) technique based on the algorithm from Niazy et al. [23] was performed to reduce this artifact on ECG data. In addition to the GA correction, the ECG presents some changes in its morphology due to the magnetic field produced by the MRI machine. Therefore, the ECG signal tends to present a T-wave larger than the QRS complex and an R-wave with reduced amplitude. Thus, traditional QRS detection algorithms tend to fail and lead to incorrect RR interval calculation.

Nevertheless, the R-peak detection algorithm proposed by Christov et al. [24] is commonly used in these scenarios, given its robustness and high performance in R-peak detection on ECG signals recorded inside an MRI scanner. The data were visually inspected to assess the quality of the R-peak detection process. After having the R-peaks detected, we proceeded to the RR interval computation to obtain the HRV time series.

### 3.4. Feature Extraction

In order to carry out HRV analysis, following pre-processing and ECG segmentation, we proceeded with the feature extraction from the code inspection data collected during each subject run. A total of 31 features across time, geometrical, non-linear and frequency domains were extracted using a sliding window of variable size and a jumping step of 1 second. The sliding window size used ranged from 3 min to 10 s, being iteratively reduced by 10 s, amounting to a total of 18 different windows (see Figure 3). All of the 31 features were extracted applying the 18 different sliding windows.

The described procedure produced vectors of individual measurements from the HRV data collected during the code inspection task (to facilitate referencing, we will call these the “extracted feature vectors”). Each individual measurement was computed based on a RR signal portion with the size of the sliding window employed. The individual measurements were then associated with the time instant corresponding to the center of the RR signal portion used to compute the respective individual measurement.

It is important to mention that the same RR signal produces “extracted feature vectors” of different lengths according to the time frame applied in the extraction process. The vector obtained with the 180-second sliding window is the one with fewer individual measurements, while the vector extracted with the 10-second sliding window is the larger, having more individual measurements. In this study, the “extracted feature vectors” using the 180-second sliding window were used as the gold standard in the statistical and correlation analyses. 

The 31 features explored in this study included six features from the time domain, three from the geometrical domain, six from the non-linear domain and 16 from the frequency domain. The different features were selected based on the current literature on ultra-short-term HRV measurements and are the result of a search conducted for the most reliable features extracted using small time frames (see Table 1).
sensors-22-06528-t001_Table 1Table 1Set of Features used in the current study presenting the designation used across the document, the units of measurement, a description of the feature and the papers reporting that feature for the analysis of HRV.HRV Features InitialsUnitsHRV Features DescriptionReferences**Time Domain**mNN [ms]mean of NN (or RR) intervals [2]SDNN [ms]standard deviation of NN (or RR) intervals [2,10]SDSD [ms]standard deviation of the differences between heartbeats [2,10]RMSSD [ms]the root mean square of the differences between heartbeats [2,10]NN50--number of consecutive RR intervals differing by more than 50 milliseconds [2,10]pNN50 [%]proportion of consecutive RR intervals differing by more than 50 milliseconds [2,10]**Geometrical Domain**TI--HRV Triangular Index–integral of the NN interval histogram divided by the height of the histogram [2,10,25,26]TINN--Triangular Interpolation of RR (or NN interval) Histogram—baseline width of the NN interval histogram [2,10,25,26]SI--Baevsky’s Stress Index [27]**Non-Linear Domain**ApEn--Approximate Entropy—measures the complexity or irregularity of the RR series [28]SD1 [ms]Standard Deviation of the Poincare plot perpendicular to the line of identity [2,6]SD2 [ms]Standard Deviation of the Poincare plot along the line of identity [2,6]PTM--Point Transition Measure—quantifies the temporal variation at the point-to-point level of the Poincare plot [6]KFD--Katz Fractal Dimension [29]HFD--Higuchi Fractal Dimension [30]**Frequency Domain**VLF [ms^2^]Very-Low-Frequency band power (≤0.04 Hz) [2,10]LF [ms^2^]Low-Frequency band power (0.04–0.15 Hz) [2,10]HF [ms^2^]High-Frequency band power (0.15–0.4 Hz) [2,10]VLFnun.u.VLF power normalized [2,10]Lfnun.u.LF power normalized [2,10]HFnun.u.HF power normalized [2,10]VLFpeak [ms^2^]VLF power frequency peak [2,10]LFpeak [ms^2^]LF power frequency peak [2,10]HFpeak [ms^2^]HF power frequency peak [2,10]VLFpeak-nun.u.VLF power frequency peak normalized [2,10]LFpeak-nun.u.LF power frequency peak normalized [2,10]HFpeak-nun.u.HF power frequency peak normalized [2,10]totPow [ms^2^]Total Power [2,10]Peak [ms^2^]Overall frequency power peak [2,10]LF/HF--Ratio of LF and HF band powers [2,10]LFpeak/HFpeak--Ratio of LF and HF band power frequency peak [2,10]

### 3.5. Statistical Analysis

In order to determine if the features extracted follow a normal distribution, the Kolmogorov–Smirnov test was performed individually by measurement in each experiment run. The test’s null hypothesis was that the data follow a standard normal distribution. At a 5% significance level, we obtained the rejection of the null hypothesis for every measurement in all runs. The conclusion was that our data do not follow a standard normal distribution, so the statistical significance and correlation tests applied must be non-parametric. Figure 4 represents the general flow chart of the experimental steps followed to evaluate the ultra-short-term HRV measurements’ reliability.

#### 3.5.1. Statistical Significance Test

To assess the sliding window size stability limit for each feature, i.e., to assess the smallest time frame that enables feature stability (when compared to the chosen reference), the Wilcoxon rank sum test was performed. In this test, the measurements extracted using the different time frames were placed against the measurements obtained using the 180-second sliding window. The test was performed independently for every experimental run and to all 31 features in the study. With the explained procedure, we were able to inspect how the variation of the window size in the feature extraction process affected the different measurements, assuming the 180-second window as a reference. 

From this process, a *p*-value was obtained for all 31 features extracted using the 18 different sliding windows, totalizing a matrix of 31 × 18 *p*-values for each experimental run of the different subjects. To analyze the global extension of this test across the different runs, we computed the percentage of runs where each feature extracted with specific sliding window size and the same feature extracted using the 180-second sliding window do not present significant statistical differences. These percentages were arranged in tables where the effect of reducing the sliding window size on the extraction can be observed (see Figure 5 and Figure 6).

In order to further investigate the time frame reduction effect, a graph was created for the results obtained from each feature. The 18 different time frames (window sizes) were placed as the independent variable on the *xx* axis, and the percentage of runs without significant statistical differences (results in Figure 5 and Figure 6) as the dependent variable on the *yy* axis. It is essential to mention that the 180-second time frame is positioned at the origin of the *xx* axis, and each unit of this axis corresponds to a reduction of 10 s in the time frame used, the minimum (10-second time frame) being the last result on the *xx* axis. The *yy* axis ranges between 0 and 100%. A linear regression was performed for the results of each feature, and the respective coefficients of determination (R^2^) were computed for each linear regression (see Figure 7 and Figure 8).

#### 3.5.2. Correlation Test

In order to complement the insight obtained with the significance test, Spearman’s correlation test was also performed. Through the application of the Spearman’s correlation test, both a *p*-value and a correlation coefficient were obtained. The *p*-value was used to determine if a significant correlation exists between the data compared, while the correlation coefficient is a measure of how correlated they are. With this test, the measurements obtained with the different sliding windows were compared to the measurements acquired with the 180-second reference time frame. Again, the procedure was conducted independently for each run. 

An important difference between the Spearman’s correlation test and the significance test of the previous section is that, in this correlation test, we must compare two vectors with the same length. As explained in the feature extraction section, extracting a measurement from an RR signal with a sliding window of 180 s produces a vector inferior in length when compared to extracting the same measurement using a sliding window of an inferior time span. Hence, with this test, we used the portion of the extracted feature vectors, computed using the smaller time frames, corresponding to the instances of the measurements in the vector resulting from the 180-second extraction (see Figure 9).

After the completion of the correlation test, for each experimental run of the different subjects, a *p*-value and a correlation coefficient were computed for all 31 features extracted using the 18 different sliding windows in the study (31 × 18 matrix of *p*-values and 31 × 18 matrix of correlation coefficients). After this step, we computed the percentages of runs where significant correlation existed, and these percentages were arranged in a matrix. The matrix lines correspond to the features and the columns to the sliding window size used in their extraction.

Regarding the correlation coefficients obtained through this procedure, we calculated the means of these values across the different runs. Due to existing runs with different sizes, the means were computed using Fisher’s z transformation. This method allowed us to give more weight to the features extracted in runs with a larger time length [31]. With this step, the average correlation across runs was obtained between the 31 features extracted with the 180-second window and the same 31 features extracted with the other time frames in the study. The Fisher’s mean values were placed in tables where the lines correspond to the features and the columns to the sliding window sizes used in their extraction. In these tables we can efficiently observe the overall effect of reducing the sliding window time span on the correlation values (see Figure 10 and Figure 11).

Similar to the process performed in the previous subsection, a linear regression was performed with the mean correlation results obtained for each feature. The coefficients of determination (R^2^) associated with the linear regression were also computed (see Figure 12 and Figure 13).

#### 3.5.3. Bland–Altman Plots

Following the recommendations of Pechia et al. [2] and Shaffer et al. [5], we proceeded with Bland–Altman plot analysis to evaluate the features’ degree of bias. A relevant difference between our approach and that of existing studies was that we performed an intra-subject and intra-run feature analysis, i.e., we tested the features’ correlation and statistical differences using different time frames within the same experimental run. In order to maintain the same intra-group analysis approach, we performed a Bland–Altman plot analysis for non-parametric data for each feature extracted with the different extracting window sizes compared with the same feature extracted with the 180-second window. This procedure was repeated for every experimental run. Figure 14 illustrates 5 Bland–Altman plot examples where it is possible to observe the level of agreement between the compared measurements.

## 4. Results

### 4.1. Statistical Significance Test Results

Figure 5 and Figure 6 summarize the results obtained using the procedure introduced in Section 3.5.1. In particular, Figure 5 summarizes the results related to the time, geometrical and non-linear domain features, whereas Figure 6 presents the results achieved using the frequency domain features. The values of each cell correspond to the percentage of runs where there is no significant difference between the feature (row) extracted using a respective window size (column), and the same feature obtained using the 180-second reference sliding window. Figure 7 and Figure 8 are graphical representations of the linear regressions obtained for the time, geometrical and non-linear domain features results and for the frequency domain features results, respectively. The feature lines chosen to be presented were the ones considered representative of the overall results. In Appendix B, it is possible to consult the slopes, the yy interceptions and the coefficients of determination obtained for every feature in the study. 

From both tables, it is possible to observe that reducing the sliding window size has a great impact on the results of the significance test in almost every feature. This drop represents a large decrease, through the time frame reduction, in the percentage of runs where there is no significant difference between extracting features using that window and the 180-second reference window. The linear regressions obtained provide quantitative and visual support for this claim.

### 4.2. Correlation Test Results

Figure 10 and Figure 11 introduce the correlation testing results described in Section 3.5.2, i.e., Figure 10 corresponds to the correlation analysis of the time, geometrical and non-linear domain features, whereas Figure 11 corresponds to the correlation analysis of the frequency domain features. Each cell value corresponds to the Fisher’s means, across the different experimental runs, of the correlations between the feature (row) extracted using a window size (column) and the same feature obtained using the 180-second sliding window. The heatmap colors correspond to the percentage of runs where a significant correlation (α = 0.05) exists between the feature (line) extracted using a given window size (column) and the same feature obtained using the 180-second sliding window. 

Figure 12 and Figure 13 allow a visual inspection of the linear regressions obtained for the time, geometrical and non-linear domain features’ correlation means and for the frequency domain features’ correlation means, respectively. Following the same scheme as Section 4.1 (statistical significance), we selected a few representative linear regression examples to be graphically presented. Appendix C contains the values obtained for the slopes, the yy interceptions and the coefficients of determination.

### 4.3. Bland–Altman Plots Results

Figure 14 depicts the Bland–Altman plots achieved for the feature LF/HF, extracted with the 120-, 90-, 60-, 30- and 10-second time frames compared to the respective features extracted using a 180-second window. The data used to perform these plots correspond to a single experimental run of an individual subject. The Bland–Altman plots allow us to observe the degree of bias present between the compared features and if the data dispersion remains within the 95% line of agreement.

## 5. Discussion

Regarding the statistical analysis and the time domain features, it is observed that four features have significance levels remaining relatively stable throughout variation in the sliding window duration, having a yy interception value close to 100 and a relatively smooth slope. The mentioned features are the SDSD, the RMSSD (basically the normalized version of the SDSD, which explains the similar percentages obtained in both measurements), the pNN50 and the mean NN (mNN). The latter two correspond to the features that exhibit the highest overall stability in the significance study. It is important to note that the linear regression obtained from the pNN50 significance results has a low R^2^ value (0.50). However, this low value results from a clear outlier in the 10-second time frame.

Keeping our attention on the non-linear and geometrical domain features, these groups have the lowest percentages of runs without significant differences between the compared features. In some cases, linear regressions with yy interception values are much further away from 100 (ApEn, KFD, SI, TI, TINN); in other cases, with very sharp slopes (SD1, SD2, HFD); or with both of these characteristics. This was an expected observation, considering the literature regarding similar studies on ultra-short-term HRV measurements. However, the point transition measure (PTM) shows promising results, since the yy interception is 94.66% and the slope is −3.04, which is a relatively soft slope in the overall context. The fact that this feature proposed in the work by Zubair et al. [6] attempts to quantify the temporal variation in the Poincare plot’s point-to-point level may help to explain the much better significance results when compared to other non-linear measurements. 

Lastly, regarding the features of the frequency domain, we notice that the features corresponding to the very-low-frequency band have the worst performance in this test. All these features have linear regression yy interception values between 70 and 80% and slope values below −5, which is a relatively sharp slope, considering other features. This result is expected, considering the current literature. It may be explained by the fact that the VLF band includes waves with 25-second periods and above (frequencies under 0.04 Hz), which means that with a sliding window of fewer than 25 s, we cannot capture a full wave, which increases uncertainty. This complication may also be extended to the low-frequency Band. Considering the scope of the LF band, whose periods range from 6.7 s to 25 s, with a 10-second or a 20-second time frame, it is not possible to capture a complete oscillation period. Even the LF band may not be the best method regarding ultra-short-term HRV measurements. This is well-reflected in the obtained results. According to Pechia et al. [2], it is recommended that spectral analyses are performed on stationary recordings lasting at least ten times longer than the slower significant signal oscillation period. This may help to explain the quick drop in the acceptance percentage results. In fact, from the 110-second window, we soon observe that most features do not have even 50% of the runs without significant differences compared to the 180-second reference extraction window. 

The results obtained based on the Wilcoxon rank sum test should be carefully analyzed due to some features’ properties and the test’s characteristics, which compare the sample distribution and its medians. In fact, a few features in the study rely on the window size used on the extraction to be computed, directly or indirectly, which affects its median values across the time frame reduction, and significant statistical differences will be found comparing these features extracted with two differently sized windows. One example is the NN50, where a larger window will naturally catch a greater number of consecutive RR intervals differing by more than 50 milliseconds for the same cognitive state. The features from the frequency domain, which compute the total power, are other examples (more oversized windows will, expectedly, have higher total power values for the same cognitive state). Furthermore, the HFD relies on the parameter “kmax” for its computation, dependent on the window size employed. On the other hand, features which are normalized values, such as pNN50, tend to have more consistent acceptance percentages through window size reduction. In this way, a comparison using an isolated statistical significance test such as the two-sided Wilcoxon rank sum, which compares the features’ medians, may lead to biased results in these features. Furthermore, we believe that having different medians does not mean the feature is not suitable for extraction with smaller windows, considering our current goal, i.e., finding the smallest time frame where each feature behavior is still representative of the corresponding 180-second measurement, for cognitive stress level discrimination purposes. 

Another problem with the isolated use of the significance analysis was that, in many features, the acceptance percentage decreases to zero very early as the window size decreases. This fact gives the false impression that, for instance, in the TI features, using a 60-second or a 10-second window essentially produces equivalent results. In this way, the correlation results corroborate some considerations made previously during the analysis of the significance tables. In addition, the linear regression obtained for the correlation results has more solid fits, having no R^2^ values under 0.90, allowing more accurate conclusions. The correlation analysis may give us more insight into how a feature changes with the reduction of the window size and, in this way, may help us to evaluate each window size until a particular feature remains reliable in our study conditions.

Regarding the correlation analysis, starting with the time domain HRV feature set, the mNN is the only feature where the correlation mean remains above 0.50 until the 60-second window (more precisely, its correlation mean remains above 0.50 until the 30-second time frame). This feature achieved the highest correlation in the smaller time frame in the study (0.40). From the literature, some studies concluded that the mNN is reliable until the 10-second time frame, such as the study by Salahuddin et al. [8], so the expectation was to see higher correlation values until smaller extraction time frames. The same can be said regarding the RMSSD and the pNN50 features. In the literature, these features are often mentioned as being reliable using 60-second time frames and lower [11,16], yet, in the current experiment, the correlation means obtained for these features using the 60-second window were already below 0.50. However, in the study performed by Salahuddin et al. [8], some recommendations by Pechia et al. [2] and by Shaffer et al. [5] are not adopted, such as the recommendations regarding the features’ bias quantification, which may increase with time frame reduction. Further, the mentioned study used a 150-second reference for the statistical analysis, while we used a 180-second time frame as a reference. Furthermore, the existing investigations, such as the study by Baek et al. [16], perform an inter-subject analysis of the features. This fact can lead to biased correlation values since it captures the inter-subject feature tendencies that may overwhelm the actual feature tendencies and increase the correlation of the features. In our present study, we perform an intra-subject and intra-run feature analysis, avoiding this kind of bias. This analysis difference explains the lower correlations obtained in the present study. It is also important to underline that, contrary to the most existing literature on the topic, our study is projected in a real-life, non-controlled environment, emulating contexts for real applications, such as bug detection algorithms based on the features under study. Accounting for these considerations, we cannot expect as high correlation values or as clean and clear results as those obtained in more controlled and resting environments, requiring lower cognitive effort.

Regarding the non-linear and geometrical domain HRV features, some features would probably be overlooked considering only the significance results. Let us take, as an example, the significance values of the KFD, the SI, the TI and the TINN. The acceptance percentages in the significance test drop to very low values in the 170-, 140-, 130- and 120-second windows, respectively. However, in the correlation results, we can observe the existence of correlation until smaller windows. Actually, in the KFD feature, more than 50% correlation is observed until 80 s, and this feature has correlation values similar to HFD. This similarity is expected since both features compute the fractal dimension. If the significance test results were the only ones taken into consideration, we could have erroneously concluded that these features are very distinct. From the geometrical and non-linear domains, the PTM was the feature that had a higher correlation within the smaller windows and with the softer slope (−0.037) of these two groups, which corroborates previous considerations.

Globally speaking, the features from the frequency domain are the ones that exhibit higher consistent correlation values for smaller windows. Several features from this domain have mean correlation values above 50% until windows of 40 s, with the HF obtaining more than 50% correlation when using the 30-second sliding window. This observation is substantially different from the significance results, which could biasedly suggest that the time domain’s features are more reliable in smaller time frames. The set of features HF, LF, LFpeak and totPow contains the features with the most promising correlation results from this domain, having correlation mean values greater than 25% at the 10-second time frame. Analyzing the slopes of the linear regressions achieved using the correlation means, it is observable that the frequency domain features exhibit higher yy interception values, maintaining a relatively softer slope. These facts indicate that their tendencies are less impacted by sliding window size reduction. Once more, as expected, the VLF band had the poorest results from the frequency domain set of features, with the steepest linear regression slopes, despite the correlation results not being as low as the literature would suggest until the 60-second window, compared to the other measurements. The set of features with yy interception values of at least 0.95 and with softer slopes of the overall study were: the mNN (a = −0.033), the HF (a = −0.038), the LF (a = −0.039), the LFpeak (a = −0.040) and the totPow (a = −0.040). These features are also the those with the higher correlation means in the 10-second time frame. Both these indicators can mean that this set of measurements is adequate to perform the intended analysis in a code inspection context.

Table 2 contains the summarized top five features by sliding window, according to the correlation mean values obtained, where one observes that the features from the frequency domain clearly stand out. In fact, only one feature from the time domain reached these top five features: the mNN when the extracting sliding window was 60 s or under, being the feature with the highest correlation when using the 10-second sliding window. The HF is the most consistent feature with the highest correlation values until the 30-second time frame.

From the significance and correlation analyses performed, it is observable that every HRV measurement present in the current study is affected by the time frame size used in their extraction. The Bland–Altman plots further corroborate this statement. These plots allow us to observe the generalized increase in the lines of agreement values of the features with decreasing extracting window size. Figure 14 corresponds to the Bland–Altman plots of the LF/HF feature extracted with 120-, 90-, 60-, 30- and 10-second time frames compared to the same measurements extracted with the 180-second time frame. In these illustrative plots, it is possible to observe this increase in the lines of agreement values with the decrease in the sliding window duration. The number of measurements that fall out of the lines of agreement also increases with reduction in time frame duration. In the LF/HF feature, this effect is very clearly observable. In this way, we can conclude that the degree of bias increases with a reduction in the analysis window size compared to the 180-second measurements. However, in this study context, it is observed that the variability which occurs in some features might be due to the fact that the samples extracted from the 180-second window capture a more overall picture of the ANS dynamics, i.e., during a window of 180-second duration, a higher degree of variability of the ANS activity might exist due to a higher degree of variability in cognitive stress during that period, in comparison to the samples extracted from the shorter windows, where a lower degree of the variability of cognitive stress is observed. This remark is in accordance with the increased variability observed as the time window duration is decreased. Therefore, given the task’s nature and application, the existence of variability on some of the Bland–Altman plots might be seen as a concern but not as a limitation for application in software engineering, given the high correlations between the two time series of comparison (180-second vs. shorter windows). These results show that the prevalent cognitive state in both windows is similar but not necessarily equal, since larger windows will capture higher cognitive state fluctuations compared to shorter windows. Furthermore, these differences can be readily captured and compensated by current machine learning and statistical techniques used to model risk scores based on HRV. 

Considering the results obtained, it is possible to observe that a set of features remains stable with the reduction in the window analysis size and is reliable for time frames of reduced duration. However, we also believe that some features that have underperformed results should not be excluded just yet, as they might contain complementary information, e.g., for class discrimination, which could be exploited when using machine learning algorithms. This should be assessed based on each problem at hand.

### Threats to Validity

In this experiment, some limitations were present, which translated into threats to our conclusion’s validity, which should also be discussed. First, it is essential to mention that the data collection study was designed with a broader goal and not specifically for HRV stability assessment, and, as such, several different biosignals and images were collected. Functional magnetic resonance imaging (fMRI) was one of the examinations performed. This examination forces the entirety of the experiment to be performed inside an fMRI scanner. The fMRI has an inherent noise effect on the ECG signal. This effect was mitigated through several ECG pre-processing and segmentation methods (Section 3.2: Pre-processing and ECG segmentation). The methods employed effectively mitigate the fMRI noise and are capable of detecting ECG peaks, which is necessary to compute a quality HRV signal. Furthermore, the subjects were alone in a quiet and isolated room when performing the tasks to control the experimental environment. Furthermore, the subjects were informed a priori about all the protocol and processes of the experiment and were instructed not to take anything that could stimulate/inhibit them the day before the experiment. Nevertheless, these external effects are minimized, given that the potential effect is blurred as we perform an intra-subject analysis, and the external effects are present in the measurements extracted with the different windows compared.

Another limitation of our study was the time frame used as a reference (180 s), which is already considered an ultra-short-term HRV. Ideally, a 5-minute (300 s) reference window would be preferable since this is a well-known and consensual time frame in the scientific community. That being said, this was not possible due to our dataset constraints. From our original 21 × 4 (subjects × runs by subject) runs, we had a few middle-run dropouts, which led to only 47 runs having more than 180 s. If the chosen reference were 300 s measurements, the dataset would be substantially reduced, leading to lower statistical power. Furthermore, the study is performed during software code inspection tasks (i.e., bug detection), which is a highly complex, dynamic and cognitively demanding task—in this study context, a 5-minute window is a considerably large window. A window of this size would capture physiological data corresponding to more than one code section, where the subject could feel different difficulty levels, leading to inaccurate results since it would capture different ANS dynamics. Another relevant constraint in the dataset is the fact that all study subjects had the same gender (male). This fact is hard to counter because the software engineering and programming fields are largely dominated by male subjects, which makes it challenging to balance the gender groups in the experiment.

Regarding the study context, most of the related work carried out until now was developed with the subjects at rest or performing elementary tasks in very controlled environments. In contrast, our study is done in a highly demanding task environment. Naturally, the dynamic characteristics of the higher cognitive function, resulting from our experimental context, will generate more dynamic signals. Additionally, the code sections inspected do not all have the same complexity. In this way, the transition from one code section to the next is expected to produce physiological signals with different characteristics and patterns, which are expected to present high variability in these periods, impacting the statistical and correlation analyses.

## 6. Conclusions

This paper studied the impact of reducing the duration of time frames on the HRV feature extraction process. The main goal of the present work was to investigate ultra-short-term HRV features to determine whether HRV-based tools can effectively be used in software development environments. To this extent, our present study investigated the smallest time frame, i.e., the finest time resolution, where each feature is reliable, i.e., the smallest time frame where each feature behavior is still representative of the corresponding 180-second measurement, under our experimental context.

Considering the results obtained, it is observed that the chosen time frame significantly impacts every feature in the study. The features from the frequency domain are those that maintain higher correlation levels until the smaller extraction window durations. From the set of the considered HRV features in this analysis, 13 features had at least 50% correlation when using the 60-second time frame (12 from the frequency domain and only 1, the mNN, from the time domain). The lower statistical significance results can be explained by the fact that features such as HF or LF compute the total power of the respective band. Using a window with a larger size will, expectedly, have higher total power values for the same cognitive state. Despite this fact, these features accurately represent the corresponding 180-second measurements’ behavior, as observable in the correlation results. Furthermore, for cognitive stress level discrimination purposes, we do not need an exact surrogate of the short-term measurements, and the feature behavior and tendencies resultant from autonomous nervous system changes can be used to evaluate different cognitive stress levels.

Regarding the smaller window size in the study (10 s), only three features exhibited at least 30% correlation: the mNN, the LF and the LFpeak. Thus, a 10-second time frame is too optimistic in our study context (high cognitive stress). The 30-second time frame is the smallest window with features with at least 50% correlation, and only two fulfilled this criterion: the HF and the mNN. The mNN, the HF, the LF, the LFpeak and the totPow features presented softer linear regression slopes of the overall correlation analysis, with a yy interception value above 0.95, meaning they are less impacted by a reduction in the time frame duration. In this way, this set of five features has shown to be the most reliable for the smallest time frames considering the present context. The mNN feature has proven to be particularly robust to the reduction of the extracting window duration. This feature had a correlation mean of 50% using a 30-second window and showed no significant statistical differences in more than 50% of the experimental runs using all the sliding windows under study, while maintaining a low degree of bias compared to the 180-second reference. 

Considering all the results, in a cognitively demanding task context, a classifier built with features extracted using time frames under 30 s might lead to inconsistent results, with potentially low scores and high deviations. However, further study is required to assess whether to discard features extracted using smaller time frames in machine learning contexts, since these features may catch some shorter cognitive patterns that larger time frames may not be able to discriminate. An approach using classifiers trained with datasets, each composed of features extracted with a different time frame, may offer more extensive insight and help to answer the raised hypothesis.

## Figures and Tables

**Figure 1 sensors-22-06528-f001:**
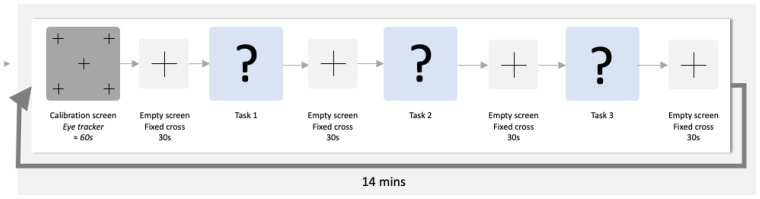
Schematic representation of an experiment run.

**Figure 2 sensors-22-06528-f002:**
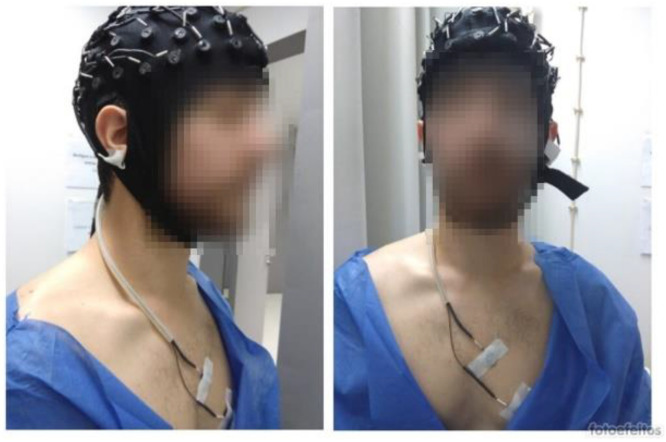
Equipment set-up used in the experiment.

**Figure 3 sensors-22-06528-f003:**
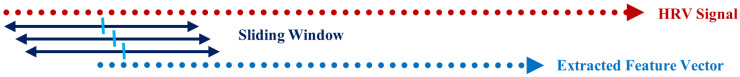
Schematic representation of the extraction of a feature using one of the sliding windows. In the end, we obtained a total of 558 feature vectors, corresponding to the 31 features times 18 window sizes for each experiment run.

**Figure 4 sensors-22-06528-f004:**
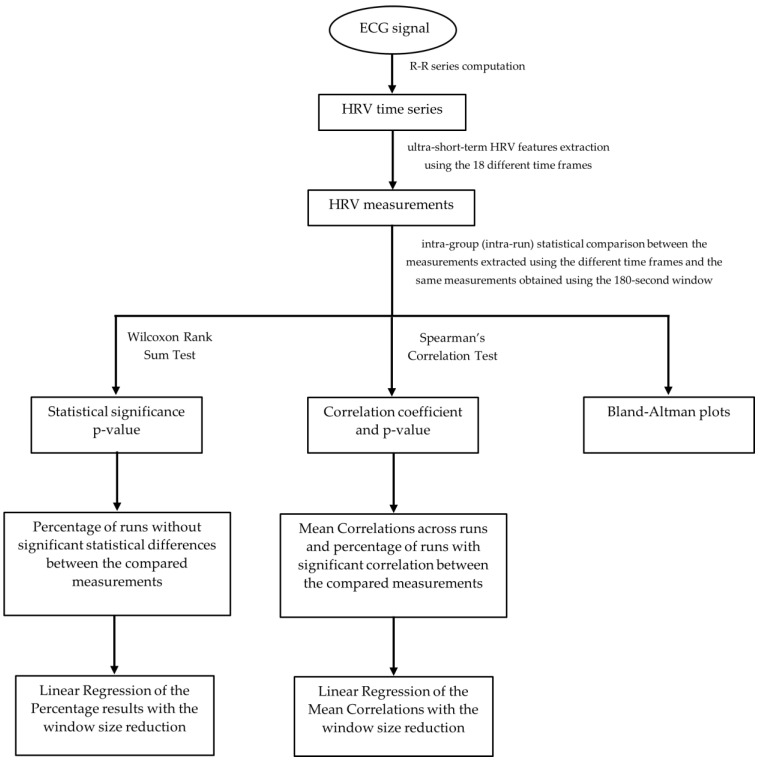
General Flow Chart of the experimental steps followed to evaluate the ultra-short-term HRV measurements’ reliability.

**Figure 5 sensors-22-06528-f005:**
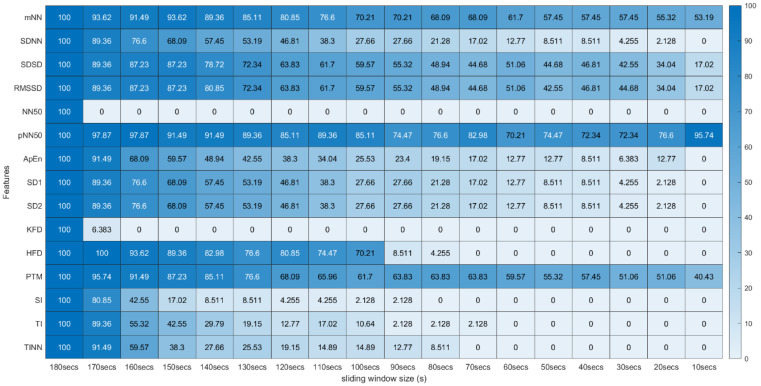
Wilcoxon Rank Sum Test (Time, Non-Linear and Geometrical Domain) * Percentage of runs where the feature (line) extracted with a respective window size (column) did not present significant statistical differences compared to the same feature extracted using the 180-second window.

**Figure 6 sensors-22-06528-f006:**
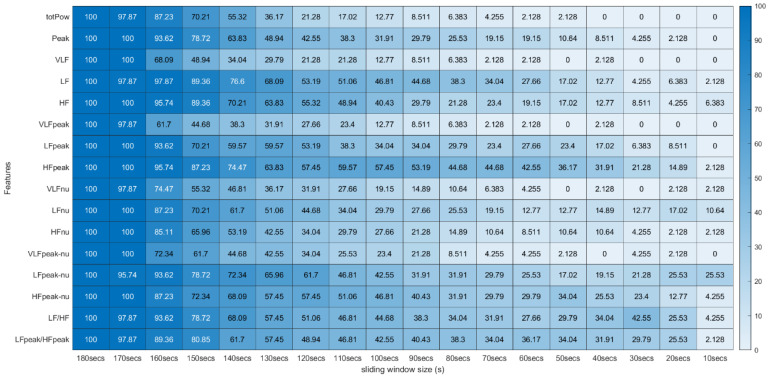
Wilcoxon Rank Sum Test (Frequency Domain) *. * Percentage of runs where the feature (line) extracted with a respective window size (column) did not present significant statistical differences compared to the same feature extracted using the 180-second window.

**Figure 7 sensors-22-06528-f007:**
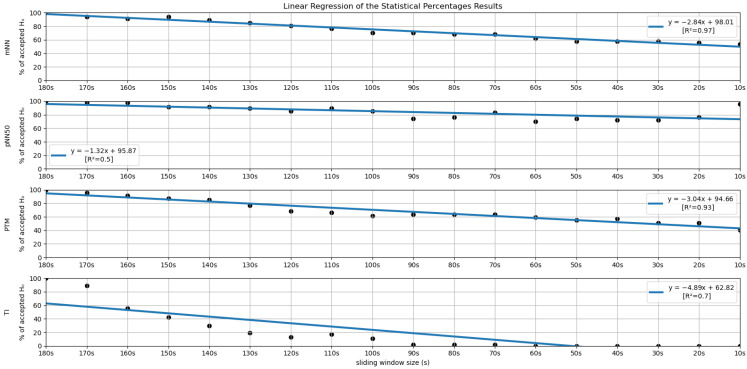
Linear Regressions of the Statistical Percentages obtained for the features mNN, pNN50, PTM and TI.

**Figure 8 sensors-22-06528-f008:**
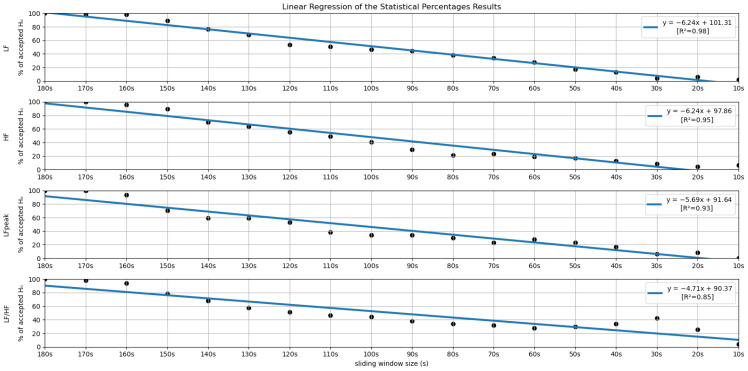
Linear Regressions of the Statistical Percentages obtained for the features LF, HF, LFpeak and LF/HF.

**Figure 9 sensors-22-06528-f009:**
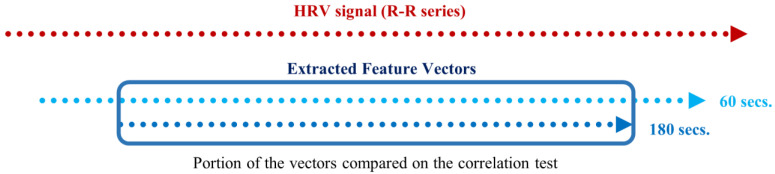
Schematic of the portion of two feature vectors compared in the correlation test, extracted using 180- and 60-second sliding windows with 1-second steps.

**Figure 10 sensors-22-06528-f010:**
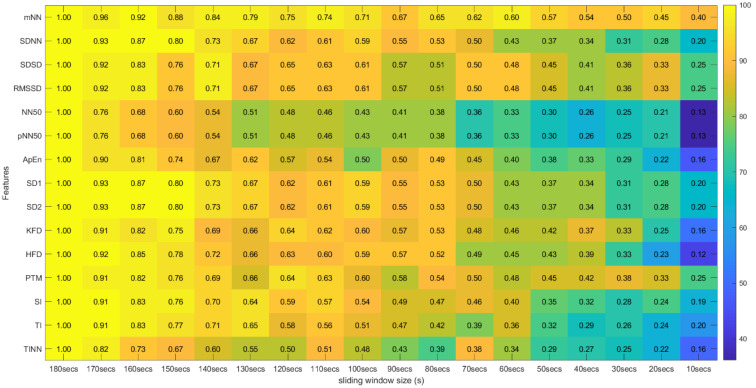
Spearman’s Correlation Test (Time, Non-Linear and Geometrical Domains) ** Heatmap Colors: Percentage of runs where there exists significant correlation between the feature (row) extracted using the respective window size (column) and the same feature obtained using the 180-second sliding window. Cell Values: Means, across the different runs, of the correlation coefficients between the feature (row) extracted using the respective window size (column) and the same feature obtained using the 180-second sliding window.

**Figure 11 sensors-22-06528-f011:**
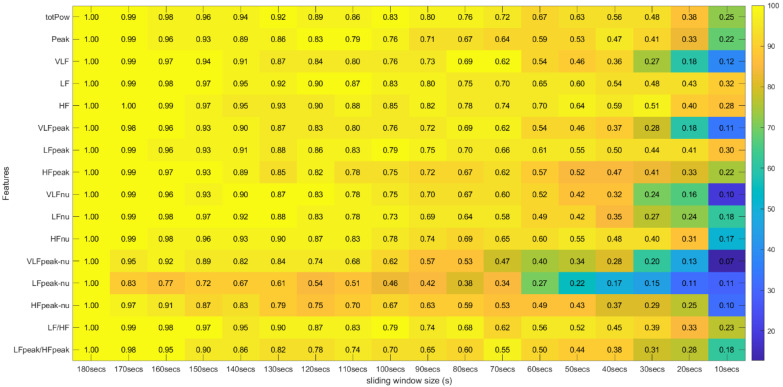
Spearman’s Correlation Test (Frequency Domain) ** Heatmap Colors: Percentage of runs where there exists significant correlation between the feature (row) extracted using the respective window size (column) and the same feature obtained using the 180-second sliding window. Cell Values: Means, across the different runs, of the correlation coefficients between the feature (row) extracted using the respective window size (column) and the same feature obtained using the 180-second sliding window.

**Figure 12 sensors-22-06528-f012:**
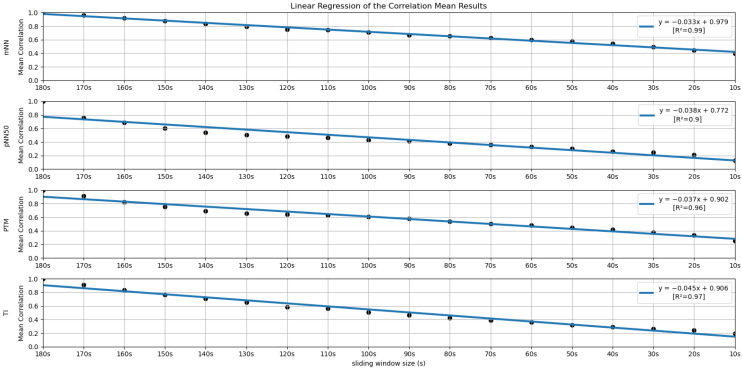
Linear Regressions of the Mean Correlations across runs obtained for the features mNN, pNN50, PTM and TI.

**Figure 13 sensors-22-06528-f013:**
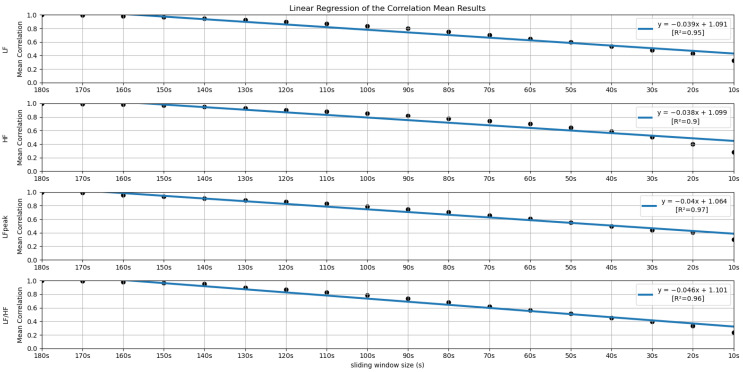
Linear Regressions of the Mean Correlations across runs obtained for the features LF, HF, LFpeak and LF/HF.

**Figure 14 sensors-22-06528-f014:**
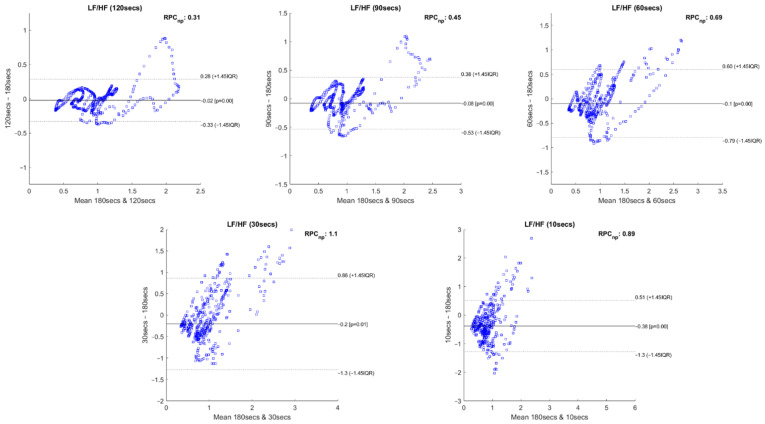
Bland–Altman plots of the LF/HF feature extracted with 120-, 90-, 60-, 30- and 10-second time frames compared to the LF/HF extracted with 180-second time frame, regarding a single experimental run of an individual subject.

**Table 2 sensors-22-06528-t002:** Top 5 features by time frame regarding the correlation means.

120 s		90 s		60 s		30 s		10 s	
HF	90%	HF	82%	HF	70%	HF	51%	mNN	40%
LF	90%	totPow	80%	totPow	67%	mNN	50%	LF	32%
totPow	89%	LF	80%	LF	65%	LF	48%	LFpeak	30%
LF/HF	87%	LFpeak	75%	LFpeak	61%	totPow	48%	HF	28%
HFnu	87%	HFnu	74%	mNN	60%	LFpeak	44%	totPow	25%

## Data Availability

The repository of the dataset used in this study is publicly available in the AI4EI platform (A European AI On Demand Platform and Ecosystem) and can be accessed in following link: https://ai4eu.dei.uc.pt/base-mental-effort-monitoring-dataset (accessed on 24 August 2022) Any other request of data can be submitted to the article authors.

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
