# Peer review of "How Reliable Are Ultra-Short-Term HRV Measurements during Cognitively Demanding Tasks?"

_sensors, 2022, doi:10.3390/s22176528_

Round 1

Reviewer 1 Report

1) This paper claims to present an assessment of small autonomic nervous system variations through ultra-short-term HRV features. The paper significantly lacks novelty and it's not clear what's new. The algorithm is not new and along with proposed techniques. Nothing is implemented on hardware as well.
2) The manuscript read like a weak algorithm paper and lacks comparison with state-of-art works. It is crucial to show the benefit of your work with prior works in the field.
3) The pain problem is not clear along with the motivation of the proposed work.
4) Abstract needs revision, please try to focus on what is new and achieved performance and also try to be quantitative.
5) (Very important) To further improve the quality of the manuscript, it is crucial to make technical comparisons with state-of-the-art works (beyond a comparison table). The methods, results, and discussion, section need to further elaborate comparison with the existing works. Some additional sample references (but not limited to these) are as follows:
.
a. M. Sohail, et al., "An ECG Processor for the Detection of Eight Cardiac Arrhythmias with Minimum False Alarms," IEEE Biomedical Circuits and Systems Conference (BioCAS), 2019. 1-4.
b. Q. Li et al., "Ventricular Fibrillation and Tachycardia Classification Using a Machine Learning Approach", IEEE Transactions on Biomedical Engineering, vol. 61, no. 6, pp. 1607-1613, Jun. 2014.
c. Z. Ebrahimi, et al., "A review on deep learning methods for ECG arrhythmia classification," Expert Systems with Applications: X, Volume 7, 2020.

6) The paper needs significant improvement in writing style. It contains redundant statements and needs proper proofreading and thorough review.

7) The conclusion should be qualitative as well as quantitative, therefore need rewriting.

Reviewer 2 Report

The manuscript presents a statistical study to compare heart rate variability (HRV) features extracted from 180-s RR interval series and shorter versions of the series obtained with a sliding window of 10-s to 180-s length. The RR series are extracted from ECGs recorded during code revision tasks. The study addresses the question of finding ultra-short term (UST) HRV features suitable for cognitive demanding tasks analysis.

The manuscript is clearly written, and the topic is clearly justified. However, results are preliminary, and the analysis and discussion need improvement before they can be published as a full research paper. The topic of UST HRV analysis is of importance and has been studied for a few years. The manuscript cites relevant works in the field and references at least two review topics [5, 9], but such references are superficial, since almost no recommendation from the cited guidelines has been adopted in the design of the present study.

Major issues:

- Line 109: best features for what? For assessing the level of stress / cognitive load…? In that case, the starting point should be to select the short-term HRV measures that can reflect that level of stress / cognitive load, according to previous studies and stablished research, and then try to find the most fitting UST surrogates.

- Following on the previous comment, the scope of the study should be better delimited. Finding surrogate UST HRV features to short-term HRV for specific applications involves several steps: (1) selecting the HRV features that capture relevant information about the ANS in the experimental situation (by comparing to control conditions, such as in stress vs. relax studies); (2) finding the UST surrogate features that most closely resemble the reference HRV features; (3) proving that the UST surrogates also capture the ANS information relevant to the experiment (for example, that they also differentiate between stress and relax conditions). The present study only addresses step (2), and the introduction and conclusions should be rewritten to clearly explain it.

- Cited works such as [13] which have been published as short conference papers may not include a thorough statistical analysis. Studies published in peer-reviewed journals such as those identified as robust /good quality in [5] and [2] could be more appropriate for comparison and for orienting the design of the present study.

- The statistical analysis lacks some of the recommended steps in the cited literature [9] and [5]. As explained in those references, non-significant differences in the median values of feature vectors do not univocally indicate whether two methods agree or have acceptable bias. Also, a significant correlation is not enough to ensure the comparability or agreement between short-term and UST measures. Bland - Altman plots should be included to quantify the bias, perhaps following the procedure for determination of "limits of agreement" in [5] or the recommended analysis flow-chart in [2].

- Results section should be separated into a Results and a Discussion section to improve legibility.

- lines 439 – 446: this hypothesis does not seem justified. The hypothesis could be partially supported if results from the present study were somehow consistent with the literature for low-demanding tasks, but no separate analysis is performed for the different type of coding task, which is a limitation of this work. Other plausible reasons of the incoherence with literature could be the use of 180-s series as surrogates for short term features, or the differences in the statistical analysis between this and cited works. Attributing the incoherence to the type of cognitive task does not seem justified by the current analysis.

- Paragraph in line 425 and line 525 in the conclusions state that correlation is a better metric to evaluate the agreement of UST and short-term features than significance testing. However, it is not correct to choose either one or the other, but both correlation and significance should be evaluated together in order to identify the USJ features with a better match. Correlation per se is of little utility [5], and the fact that significance testing does not provide the most convenient results to the study’s goals does not mean that it can be ignored when needed.

Other issues: 

- Line 39: provide references to those studies

- Line 362: references needed

- In the methods section, include the references to the tables’ number to improve legibility (for example, in the paragraph in line 262, the reference to tables 2 and 3 should be included).

- In the captions of tables, indicate briefly what the contents are (scales, colors, etc).

- Regarding the lack of female subjects, surely the number of female developers is low but not zero. According to the BSE project webpage, the study is ongoing, so perhaps recruiting efforts could be improved by designing the recruitment materials for the study in a way that is encouraging to women, providing female volunteers with extra benefits, including female researchers in the team who can be the person of contact for volunteers, etc.

Reviewer 3 Report

In this manuscript, the authors explored the reliability of the ultra-short-term HRV features during cognitive demanding tasks. The structure is disorganized and needs to be readjusted. And I have some suggestions for this paper.

1. In Abstract, the first appeared abbreviations should be given their full name, e.g. ANS. Additionally, only the experimental results are described, but the conclusions need to be supplemented in Abstract.

2. The structure of this paper should be reorganized, e.g. there is only section 3.1 in Section 3, but not section 3.2. I think the authors should reorganize the structure carefully, and I suggest the section 3.1 should be integrated in the Method or Results section. Moreover, the Discussion should be separated from Section 5, and be integrated in Section 6.

3. A general flow chart in Method is necessary, so that readers can understand the experimental steps more clearly. Figure 3 and Figure 4 could be synthesized and prettified.

4. In lines 94-103 of the Introduction, the author describes that the demand for the use of ultra-short HRV features has grown and brought some new applications. Please show more details about the necessity or application scenarios of this research under cognitive demanding tasks. 

Reviewer 4 Report

Overall this is a well written manuscript. Please see my comments below

Introduction

Line 111-112 is very confusing

On line 116, you can use the name of the authors to identify the study before citing 13.

Other than that I would recommend finding a better way to identify studies rather than stating something along the lines of "In study [#]"

Methodology

There are several methodological issues that need to be addressed to improve replicability

1. How were subjects recruited?

2. I understand that you had programming experience and technical questions regarding the inclusion/exclusion criteria however, how did you characterize programming experience? And what is the difference between technical questionnaire and programming experience? Please provide both surveys in an appendix. It would help with replicability.

3. With the larger variation in age and mean being 22, I am assuming you had a few outliers. You should provide median for the age instead of mean since the range is 19-40 with mean being 22.

4. I think in your experimental protocol, you should mention that you used a randomized control crossover design. It is the study design. Further it helps solidify your strong research design.

5. Based on the picture show, although I can't tell, but were you also collecting EEG. If you were can you please provide that information in the methodology and then identify the fact that you did not use that information in this analysis.

6. How did you control the environment in which these subjects performed these tests? Did they perform them with other people around? Were they performed individually? Did you control for caffeine and sleep the night before? Did you control for any other stimulants?

7. Great job in describing the statistical analyses. My only question is can you identify the variables that were used in the linear regression and what was the total n in each of the models. Your data size was small (n=22) and not normally distributed which means that you can't use CLT or LST to justify running a linear regression. I am wondering how you were able to run one.

Results:

The written part of the results are great however, the tables are a bit confusing.

1. I appreciate that you stated in the results section that the data presented in Tables 2 and 3 are the percentage of the runs that were not significantly different. Can you please put a legend for those tables and state that

Threats to validity

1. I think that your context of having these subjects perform cognitively demanding tasks is one of the strengths of your study. 

The conclusion is also very well written.

Just some minor concerns. There are some places where there are small grammatical errors, but other than that the writing is great. 

Round 2

Reviewer 3 Report

The problems I mentioned have been addressed.

Author Response

We would like to thank you for your constructive comments and suggestions. We do believe they have contributed to enriching the paper and making it clearer and more useful to readers.

Reviewer 4 Report

I appreciate the authors responding to my queries. I also appreicate them adding the Bland-Altman. However, based on the Bland-Altman results the authors make too large of a jump in their conclusions as the Bland-Altman clearly show that there is a high degree of variability in the results thus suggesting that none of the proposed time frames are good ways to measure HRV

Round 3

Reviewer 4 Report

Thank you for adding lines 604-620

Author Response

(The authors gave the same response as above.)
